# Under Pressure: Time Management, Self-Leadership, and the Nurse Manager

**Elizabeth Goldsby [1,*], Michael Goldsby [2], Christopher B. Neck [3] and Christopher P. Neck [4]**

[1]  School of Nursing, Ball State University, Muncie, IN 47306, USA
[2]  Department of Management, Ball State University, City, Muncie, IN 47306, USA; mgoldsby@bsu.edu
[3]  College of Human Sciences & Education, Louisiana State University, Baton Rouge, LA 70803, USA; cneck1@lsu.edu
[4]  Department of Management and Entrepreneurship, Arizona State University, Tempe, AZ 85257, USA; christopher.neck@asu.edu
[*]  Correspondence: eagoldsby@bsu.edu

**Abstract:** Decision making by nurses is complicated by the stress, chaos, and challenging demands of the work. One of the major stressors confronting nurses is perceived time pressure. Given the potential negative outcomes on nurses due to perceived time pressures, it seems logical that a nurse manager's ability to lead nurses in moderating this time pressure and in turn to make better decisions could enhance nurse well-being and performance. Paralleling research in the nursing literature suggests that, in order to improve patients' judgement of the care they received, nurse managers should embrace ways to lower nurses' perceived time pressure. In this conceptual paper, we propose a model to help mitigate time pressure on nurse managers and their frontline nurses based on the research regarding time pressure, psychosocial care, time management, and self-leadership. Three metaconjectures and suggested future studies are given for further consideration by organizational and psychological researchers.

**Keywords:** nurse manager; time pressure; self-leadership; stress

## 1. Introduction

People who agree to assume important management roles in organizations often bear many responsibilities to a varied set of stakeholders in their daily work. Given the impact managers have on their organizations, research has examined many facets of the challenging nature of this work. The better scholars can shed insight into managing and mitigating the stressors management positions hold, the better the manager will perform as both a professional and person. In turn, the better the manager performs, the better the organization will do as well (Bakker and Demerouti 2007). The job demands research tradition is one management area that holds particular significance for studying issues managers face in their daily work (Demerouti et al. 2001). Specifically, job demands have been defined as "the degree to which a given executive experiences his or her job as difficult or challenging" (Hambrick et al. 2005, p. 473). Job demands are not inherently a negative phenomenon in the workplace. If job demands are reasonably manageable, many managers may find the challenges interesting and satisfying, as the work offers them opportunity to apply and develop their expertise and experience (Gardner 1986; Garner and Cummings 1988; Janssen 2001; Scott 1996). After all, people in leadership positions usually reach their status from seeking and succeeding in situations others may avoid. However, researchers have also discovered that overly taxing job demands can entail great mental strain and stress (Karasek 1979; Van Yperen and Snijders 2000; Wall et al. 1996; Xie and Johns 1995) and incur physical health problems (Fox et al. 1993;

Theorell and Karasek 1996; Warr 1990). The nature of whether a job demand is stimulating or taxing is dependent on three factors: task challenges, performance challenges, and personal performance aspirations (Hambrick et al. 2005). The degree of stress a manager finds in addressing their tasks, organizational expectations, and personal aspirations can also impact the quality of their decision-making (Ganster 2005). Therefore, overly demanding jobs can lead to poor decisions by managers. This negative consequence is multiplied when made by leaders who supervise outcomes with great impact on their organization and/or society. Few professionals in society make more important decisions than nurse managers. They routinely are faced with "life or death" situations requiring decisions of how their frontline nurses are to proceed. Better practices for assisting nurse managers with the demands they face in their work will improve their decision making and, ultimately, will better serve their patients with quality care as well as address patient safety.

The job of the nurse is filled with much stress and chaos given the challenging demands within today's medical environment (Goldsby et al. 2020; Greggs-McQuilkin 2004). One of the major stressors confronting nurses is perceived time pressure (Teng et al. 2010). Time pressure impairs the decision making of nurses (Hahn et al. 1992), reduces their emotional well-being (Gärling et al. 2016), and leads to nurse exhaustion Gelsema et al. (2006). Furthermore, recent research in the psychological sciences suggests that increased time pressure can lead to more dishonesty (Protzko et al. 2019). Given such potential negative outcomes on nurses due to perceived time pressures, it seems logical that a nurse manager's ability to help nurses manage this time pressure and become better decision makers (that is, become better time managers) could enhance nurse well-being and performance. Research in the organizational time management literature shows a positive relationship between time management and job satisfaction, health, and performance outcomes (Claessens et al. 2007). Paralleling research in the nursing literature suggests that in order to improve the perception patients have of the quality of care they receive, nurse managers should embrace ways to lower perceived time pressure (Teng et al. 2010). However, while time pressure on nurse managers has been studied as a common problem in hospitals, a theory-based framework for better performance within those constraints has not been provided in the literature. We seek to address that literature gap by answering the following research question in the upcoming sections: How can nurse managers and nurses provide quality patient care by making better decisions under time pressure? This conceptual paper provides a step in that direction; that is, to help nurse managers reduce perceived time pressure by nurses and improve the decision making of nurse managers and nurses. In this article, we suggest how evidence-based time management practices (Garbugly 2013; Saunders 2014) and self-leadership theory (e.g., Neck et al. 2019) can help nurse managers to reduce perceived time pressure by the nurses that they manage. We capture these insights in a framework we call *The Time Pressure Mitigation Model for Nurse Managers* (see Figure 1). We propose that guidelines inherent in this model will also serve other managers who find themselves making decisions under time pressure constraints.

Our proposed model mitigates the time pressure nurse managers face based on three areas of research: psychosocial care, time management, and self-leadership. Based on the conceptual methodology of metatriangulation (Lewis and Grimes 1999; Saunders et al. 2003; Cristofaro 2020), we provide three metaconjectures for further consideration by organizational and psychological researchers. Metaconjectures are "propositions that can be interpreted from multiple paradigms" (Saunders et al. 2003, p. 251). Saunders et al. (2003), for example, applied the approach to examining power and information technology. The context of nurse managers is included in the title of the framework because the professional outcome pertinent to their roles is psychosocial care. Based on the research on nurse management studies, we conjecture that time pressure will impede good decision making and detract from providing quality psychosocial care. However, the research in time management and self-leadership warrant us to also conjecture that when practices from these two areas are successfully implemented, the negative effect of time pressure on decisions related to psychosocial care can be lessened. In other words, proper application of time management and self-leadership practices moderates the relationship between time pressure and psychosocial care by nurse managers.

In this conceptual paper, we first describe the context nurse managers face that affects their decision making. We then provide the model and its constituent parts. An example is then given that demonstrates how the model may work in a healthcare setting. We conclude with considerations for future development of the model.

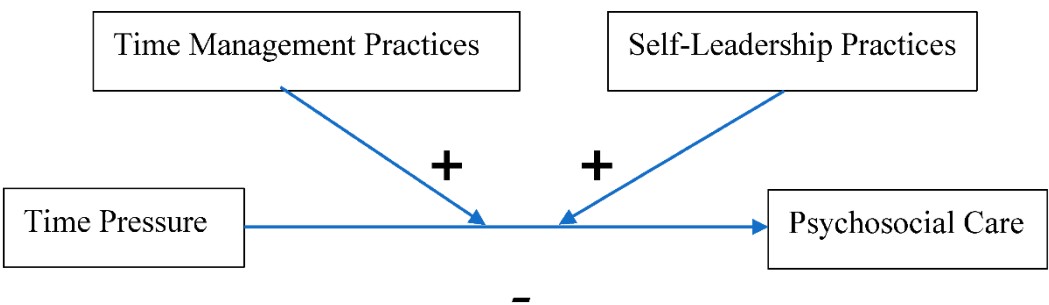

**Figure 1.** The time pressure mitigation model for nurse managers.

## 2. The Context of Healthcare Settings

Time is a major issue in healthcare today. The emphasis on quality of care, safety, standardization, and efficiency has to be managed within the constraints of an increase in the amount of patients being treated and a condensed length of stay (Bundgaard et al. 2016). Nurses are often in a continuous struggle to perform an increasing number of complex tasks under time-crunched conditions (Chan et al. 2013). Further complicating the nurse's job are the many decisions that must be made within that limited time (Saintsing et al. 2011). In a study of a medical admissions unit, it was found that a nurse confronts up to 50 important clinical decisions in a single 8-hour shift (Thompson et al. 2004). Other researchers have discovered similar patterns of clinical judgements and choices in controlled time frames. Thompson et al. (2008), for example, discovered that nurses in intensive therapy units encountered a clinical judgment or decision every 30 seconds. Along with that, in a study by Saintsing et al. (2011), nurses reported time constraints that limited their patient assessments with approximately 80% of the novice nurses acknowledging making mistakes due to time pressure. In this study, it was reported that each nurse made better decisions when there was no time pressure confronting their interactions. Additionally, Gonzalez (2004) demonstrated that people making decisions under limited time conditions performed worse than others faced with the same situations but with more time.

Accelerated information processing (Maule and Edland 1997) has been recognized as a natural response to time pressure when implementing a desired strategy (Payne et al. 1993). Furthermore, novice nurses described that peer-pressure is an indirect basis of time constraint (Ebright et al. 2004). Specifically, they feel pressure to leave no unfinished tasks for the incoming nurses that start the next shift, pressured to complete their allotted tasks so that incoming nurses start their shifts without leftover work. This self-inflicted time constraint is pervasive in healthcare, especially for new nurses who want to avoid being seen as incapable of thoroughly completing their responsibilities. The "must-do work" supersedes the "should-do work" (Bowers et al. 2001), thereby causing nurses to perform in a reactive fashion rather than being proactive with decision making (Hoffman et al. 2009). Nurses report that pressure to perform, workload, technology, and system issues produce barriers that prevent them from providing compassionate care. The often-chaotic environment weakens their capacity to care for others as well as themselves. This can result in dissatisfaction, burnout, and compassion fatigue (Roussel et al. 2020). Moreover, time pressure bears a high price of energy expenditure as nurses rely on coping mechanisms to bear the increased anxiety that comes with time pressure. In other words, nurses are not able to fully focus on their job, but must engage in self-coping to stand up to the pressure. This unfortunate predicament can drain energy over a work shift (Maule et al. 2000). Given this context of time pressure within a healthcare setting, we next

explain the Time Pressure Mitigation Model for Nurse Managers and offer related metaconjectures for future study.

## 3. The Time Pressure Mitigation Model for Nurse Managers

Time pressure is the perception that scarcity of time exists to finish obligatory tasks (Teng and Huang 2007). Nursing responsibilities have been rising along a range of complexity. This amplified workload can complicate nursing (Gurses et al. 2009). Too much to do without enough help was found to be the top source of stress in a study of Iranian nurses (Mosadeghrad 2013). In the study, occupational stress was found to lessen the quality of care due to having less time to demonstrate compassion for the patients in their care. Additionally, mistakes and practice errors occurred more often when occupational stress increased. Stress is related to time pressure in that it is recognized as inadequate time for accomplishing required tasks that compromises one's ability to cope. Under intense time pressure, it was found that individuals tend to escalate information processing, hinder decision-making quality, and experience information overload (Ben-Zur and Breznitz 1981; Hahn et al. 1992). Concurrently, the stress of time pressure has physical complications, such as increased blood pressure and a rise in human cortisone levels (Wellens and Smith 2006; Greiner et al. 2004). Thompson et al. (2008) found that time pressure reduced the nurses' capacity to assess patient needs in acute care and affected nurses' risk assessment decisions. Time pressure also creates negative emotions, increases anxiety, and leads to nurse emotional exhaustion (Gelsema et al. 2006) or burnout (Ilhan et al. 2008). When in a state of high anxiety, one's working memory resources are constrained, significantly diminishing resources for completing tasks and ultimately diminishing individual effectiveness. Patients are aware of this limited cognition, which can diminish client satisfaction and confidence in the hospital. Teng et al. (2010) discovered that nursing-perceived time pressure is negatively related to patient perceptions of dependability, accountability, responsiveness and assurance of the nurse. Thus, to enhance patient perception of care quality, nursing managers must develop means to lessen nurse-perceived time pressure.

### 3.1. Time Pressure and Psychosocial Care

Psychosocial care, a holistic approach to nursing to meet the psychological and social needs of patients (Kenny and Allenby 2013), is an important healthcare outcome compromised by insufficient time and heavy workloads (Legg 2011). Barriers, including lack of time, stand in the way of appropriate psychosocial care (Legg 2011). Recent studies reveal that offering good psychosocial care may improve patient overall health outcomes (Chen and Raingruber 2014). Additionally, appropriate psychosocial care reduces patient anxiety and stress and alleviates pain, thereby improving quality of life as well as a reduction in hospitalization cost due to a decreased need for medical resources (Kenny and Allenby 2013; Legg 2011). Studies by Legg (2011) and Rodriguez et al. (2010) found that good psychosocial care decreased the duration of hospitalization.

Unfortunately, psychosocial care does not often become an area of focus in hectic acute care settings (Legg 2011; O'Gara and Pattison 2015). In a study by Chen and Raingruber (2014), all the participants stated that, although often limited, positive interactions with patients and their family members were vital for providing psychosocial care. Communication within a time-pressured environment is hard to come by. As one nurse in the study said, "I think in order to know the needs of the patient, you need to communicate with them, and then you will know what they need." (p. 229).

The literature supports that effective interactions between nurses and patients result in increased rapport, trust and medical care, thus making therapeutic relationships possible (Belcher and Jones 2009; Josefsson 2012; McMillan et al. 2016). Furthermore, although nurses support spiritual care, they commonly said it was not possible when under time constraints (Balboni et al. 2014). Most participants in Legg's (2011) study responded that time constraints due to excessive workloads was the top obstacle to offering psychosocial care. Time constraints shortened conversations with patients that would better uncover individualized needs. Other studies support this conclusion that most nurses contend with time-related pressures (Legg 2011; Chen and Raingruber 2014; DeCola and Riggins 2010; Lawless et al. 2010). A likely cause for this occurrence are

the high patient workloads in hospitals and the distribution of tasks that require a specific schedule that needs completed before the end of one's shift (Lim et al. 2010). An enormous preoccupation with documentation also limits time available for psychosocial needs (Legg 2011). Most nurses in this study expressed that a preoccupation with timely documentation requirements limiting patient interaction was the cause of not attending to psychosocial needs.

Quality decisions within a healthcare context are also affected by time pressure. Good decisions are greatly dependent on the information considered by decision makers. However, new findings in the psychological sciences reveal that time pressure can cause nurses to misrepresent actual events and results, in order to appear more favorable to other people (Protzko et al. 2019). This was affected, though, by participants' beliefs on whether their true self was virtuous. Bear and Rand (2016) and Rand et al. (2014) proposed that one's automatic responses develop from internalizations of actions commonly agreed to as good to others in social exchanges. With this notion in mind, when time pressure is at hand, an individual might say what would appear to be the right response when indeed it might not be the truthful response. An example of this could be when a nurse reports to the nurse manager that she/he has followed protocol and checked all the necessary patient identifiers before giving medications during a busy shift, while in reality the nurse is telling the nurse manager what she/he thinks she/he wants to hear. This error in judgment could result in a medication error, which is one of the most common—and dangerous—mistakes in healthcare. When people are under time-crunched conditions, they often offer socially desirable answers or information as a default. A clash between a person's long-held self-conception and workplace role takes place. Thus, under time pressure, people often operate in opposition to their true self-concept, responding consistently with the internalized social norm of how the unit is supposed to operate from day-to-day (Everett et al. 2017). Positive self-presentation becomes a habitual tendency when time pressure is present (Protzko et al. 2019). When intentions and actual events are in misalignment, cognitive dissonance can place further stress on the nurse's psyche—sometimes lasting well beyond the date of the actual occurrence.

Given this discussion of the effect of time pressure on psychosocial care, we offer the following metaconjecture based on the literature:

**Metaconjecture 1:** *In situations where nurse managers face increased time pressure, providing quality psychosocial care will be compromised; i.e., the more time pressure a nurse manager experiences, the less psychosocial care their patients receive.*

### 3.2. Time Management for Nurse Managers

Nurse managers must create a positive work environment even when they are confronted with ever-changing priorities. They must especially consider time pressure of their staff when designing their schedules. Workload should relate to a realistic assessment of individual nurses' capabilities and resources. Adequate staff can help in keeping nurses' assignments realistic to sufficiently manage workloads (Waterworth 2003). Nurse managers should initiate strategies to provide substantial support for the nurses to deal with the stresses that are at hand. Numerous studies have considered time pressure and work overload as major contributors to work stress among healthcare professionals. A burdensome workload intensifies job tension and reduces job satisfaction which, in turn, increases the probability of turnover. Although, Efron (2014) identifies poor leadership as the main reason for staff to leave and notes that staff quit the leader, not the organization. Roussel et al. (2020) report that the highest turnover takes place within one year of employment, with the cost of replacement at USD 75,000 due to recruitment, temporary staff, overtime, orientation and replacement. Moreover, the remaining staff are affected with heavier work assignments and overtime, which also leads to burnout. If the vacancy rate remains high, burnout may lead to more vacancies and, in turn, increase the potential for further burnout among the remaining staff leading to a downward spiral. It is as if a unit is unintentionally downsizing itself.

Inadequate staffing also impacts quality of care and patient outcomes (Aiken et al. 2002). Conscientious management of all these factors is key. Time management for nurse managers is an important issue (Mirzaei et al. 2012) because it directly affects people's health, availability of critical

time, and can cause a decrease in efficiency (Soleymani et al. 2011). It was found in a study by Ziapour et al. (2015) that training nurse managers according to time management practices delivers positive results in healthcare centers.

A main objective for the nurse is to make optimum usage of the time at hand. Nurses keep part of their focus on maintaining the expectations of the greater healthcare system while also providing individualized care in the most efficient way they can. Thus, instead of developing a relationship with a patient, much of the allotted time is spent on the technical and instrumental responsibilities in nursing. As a result, compassionate nursing becomes harder to provide (Bundgaard et al. 2016). Therefore, productivity, and not compassion, becomes the key objective of the job. Numerous researchers cited Lakein (1973) when studying time management, emphasizing the practices of needs assessment, setting goals to meet these needs, prioritizing, and task planning necessary to meet the set goals. Practices proposed to extend intellectual efficiency were suggested by Britton and Tesser (1991). Kaufman-Scarborough and Lindquist (1999) provided methods for strategizing activities by prioritizing them by their relative importance to the healthcare mission.

Collections of behaviors that are considered to aid efficiency and lessen stress were suggested by Lay and Schouwenburg (1993). Based on a review of the time management literature, Claessens et al. (2007) suggest the following definition: "behaviors that aim at achieving an effective use of time while performing certain goal-directed activities" p. 262. Since the focus is on goal-directed activities that are accomplished in a manner that implies successful use of time, the following behaviors are included in their definition: (1) Time Assessment Behaviors—focusing on mindfulness of the past, present and future with self-awareness of time usage within the boundaries of one's abilities (Kaufman et al. 1991) and self-awareness of time handling by deciding which tasks suitably fit into one's abilities; (2) Planning Behaviors—with the goal of effective use of time, that includes goal setting, development, prioritizing, formulating a to-do list, and arranging tasks (Britton and Tesser 1991; Macan 1996); and (3) Monitoring Behaviors—with the objective of attending to how time is allotted, engagement of planned undertakings, and limiting the impact of disturbances by others in the completion of tasks and goals (Fox and Dwyer 1996; Zijlstra et al. 1999).

Nurses face time management problems due to the unpredictability and complexity of their assignments. Accomplishing tasks effectively and minimizing interruptions are essential to the nurse. The importance of routines and prioritization is key to time management in a healthcare setting (Waterworth 2003). Furthermore, in complex environments, routines allow for a way of maintaining order since actions have already been planned out and can decrease thinking time needed to make decisions. Routines bring about a sense of predictability, awareness of time control, and familiarity of experience that is pertinent to time management (Waterworth 2003). Furthermore, prioritization is a prerequisite for effective work performance for nurses. Sequencing work and its duration is necessary as well. It is imperative for timing and speed that there is synchronicity with others in the nursing environment. Determining what is urgent and important is a critical step to attaining high returns on time investments. Simply put, more hours worked does not mean more hours of productivity. Therefore, productivity experts offer many suggestions for being more deliberate and conscious of where time on activities is allocated. Some of the latest advice for nurses on time management includes (Garbugly 2013; Saunders 2014):

1.  Never relying solely on your memory and instead referring to reminders and lists.
2.  Accomplishing the most important task as early in the day as possible.
3.  Paying attention to the time of day that you are most productive and utilizing that time for your most important tasks.
4.  Keep multitasking to a minimum. Many psychologists believe that multitasking does not actually exist, meaning you can only put your attention on one thing at a time. When people think they are multitasking, they are actually only shifting their attention inefficiently from one matter to another in quick bursts. Each time a person moves their attention back to a previous matter, a transition in cognition must take place. Any momentum the person had in their thought process is interrupted, and the brain must reorient to the new focus. These reorientations may be minute, but over the course of hours, days, and weeks, significant time

can be lost in perceived "multitasking." Thus, it is more efficient and productive to complete tasks with full attention and then move onto the next one needing accomplished.

5.  Attending to emails only at set times each day, and, when possible, for a determined amount of time.
6.  Keeping your work area neat and organized. It can help minimize search time for needed resources. Additionally, many productivity experts believe that removing clutter in a physical space helps the mind to focus attention more fully on that matter at hand.
7.  If able, finishing small tasks before handling larger ones.
8.  Defining what work needs to be done the next day and writing it down before the end of the shift.
9.  Taking breaks and doing something enjoyable after you have accomplished a task. Recharge a bit, if possible, before moving onto the next task that needs attention. Improved productivity is a long-term game, not a short burst of frantic task hopping.
10. Enjoying the dopamine that the brain secretes when tasks and goals are accomplished. Completing activities feels good and serves to encourage further accomplishment. Therefore, consciously managing activities and the time required for their accomplishment boosts mental and physical health by releasing positive neurochemicals into the bloodstream, as opposed to excessive cortisol that is released over time in unorganized and pressure-packed environments (Lee et al. 2015).

Given this extensive review of research on time management theory and practices for better performance, we offer the following metaconjecture:

**Metaconjecture 2:** *In situations where nurse managers face increased time pressure, proper application of research-based time management practices can improve psychosocial care; i.e., time management practices positively moderate the negative relationship between time pressure and psychosocial care.*

*3.3. Self-Leadership for Nurse Managers*

The research in self-leadership suggests that it can be an appropriate training tool for nurse managers in better performing their roles. Based in social cognitive theory, self-leadership can help nurse managers better manage their thoughts, behaviors, and environment to create a better workplace for improved results. Self-leadership (Manz 1986; Manz and Neck 2004) is a process in which people can regulate what they do, how they interact with others, and how they decide to lead themselves and others by using certain behavioral and cognitive strategies. Self-leadership strategies fall into three groups focused on behavior, natural rewards, and positive thought patterns (Manz and Neck 2004; Prussia et al. 1998; Neck and Houghton 2006). Strategies revolving around behavior improve the awareness a person has on what they are trying to accomplish, especially regarding tasks with which one might want to procrastinate (Manz and Neck 2004; Neck and Houghton 2006). Behavior-focused strategies are:

1.  Self-observation—Developing the self-knowledge of when and why a person participates in the actions she/he does. In the context of nurse managers, this suggests that the self-awareness of the antecedents and consequences of perceived time pressure is critical. Self-awareness is a crucial aspect of altering or eradicating self-destructive or limiting behaviors; (Manz and Sims 1980; Manz and Neck 2004; Neck and Houghton 2006).
2.  Self-goal setting—Having awareness of present actions and results can help a person set meaningful goals for themselves (Manz 1986; Manz and Neck 2004; Manz and Sims 1980; Neck and Houghton 2006). Research supports the effectiveness of establishing challenging and precise goals to improve a person's performance (Locke and Latham 1990; Neck and Houghton 2006).
3.  Self-reward—Personal goals that are met with rewards one finds pleasing and desirable can encourage a person to take the initiative to overcome procrastination and/or poor prioritization (Manz and Sims 1980; Manz and Neck 2004).
4.  Self-punishment (also known as "self-correcting feedback")—Entails positive honesty, reframing failures and unproductive actions in a way that can help a person remodel future

actions. This strategy comes with a caveat, though: self-punishment centered on self-criticism should be used sparingly, lest a person incur excessive guilt that damages self-esteem, self-efficacy, and self-confidence that hinders future performance (Manz and Sims 1991; Neck and Houghton 2006).

5. Self-cueing—Designing your work environment with reminders to maintain positive self-leadership behaviors and thoughts. Concrete environmental cues such as notes, lists, and inspiring quotes can help a person return their attention to making progress toward their goals. For example, nurse managers could place pictures in the rooms in which they work reminding them to take deep breaths and focus on the patients on the unit at that particular point in time.

Natural reward strategies are designed to establish conditions that spur correct actions through focusing on the gratifying aspects of a task (Manz and Neck 2004; Neck and Houghton 2006). These strategies encourage a sense of competence and self-determination in the person practicing them, two key drivers of intrinsic motivation (Deci and Ryan 1985). The necessity for competence comprises the need to practice and increase a person's proficiencies, and self-determination implicates one's desire to be independent from pressures such as conditional rewards. When individuals feel negatively controlled by their environment and they associate their expected actions to external pressures, they are likely to be less motivated by the work itself. To avoid this negative perspective of work, two natural rewards strategies that can be practiced are:

1. Building positive features into an activity, so that doing it becomes a reward in itself (Manz and Neck 2004; Manz and Sims 1991). For example, if a nurse manager likes music, she/he could relate what she/he wants to accomplish on the unit at the moment with a song. Perhaps she/he could sing to himself, "Everybody's workin' for the weekend!" as she/he looks at timesheets.
2. Deliberately turning attention from the ungratifying features of a task and placing it on the more inherently rewarding characteristics of the required action (Manz and Neck 2004; Manz and Sims 1991; Neck and Houghton 2006). An example for the nurse manager could be a daily mental reminder to themselves and their staff as to why they entered the profession in the first place—that is, a reminder to help and care for people. This reminder could help the nurse manager focus on the naturally rewarding aspect of the job instead of focusing on the perceived time pressure.

Effective thought pattern strategies are devised to enable a positive stream of recurring thoughts and construction thinking habits that can enhance a person's performance (Manz and Neck 2004; Neck and Manz 1992). Positive thought pattern strategies include:

1. Acknowledging and replacing dysfunctional beliefs and assumptions—A person should scrutinize thoughts that are not helpful to achieving goals and exchange them for more rational and productive thoughts and beliefs (Ellis 1977; Manz and Neck 2004; Neck and Manz 1992).
2. Practicing positive self-talk—What we quietly say to ourselves should be positive (Neck and Manz 1992, 1996a), including our self-evaluations and reactions to events (Ellis 1977; Neck and Manz 1992). Negative and unhelpful self-talk should be acknowledged and exchanged with helpful internal monologues. Mindfully observing the patterns we use to talk to ourselves helps us to replace unconstructive self-talk when it arises. The mind can only focus on one matter at a time, so it is better to place its attention on self-dialogues that are optimistic and hopeful (Seligman 1991).
3. Practicing mental imagery or visualization—Develop the skill of intentionally imagining a future event or task in advance of its actual occurrence (Finke 1989; Neck and Manz 1992, 1996a). Those who can picture successful completion of a future event or task before it is actually performed are more likely to attain that result (Manz and Neck 2004). Moreover Driskell et al. (1994) conducted a meta-analysis of 35 empirical studies and discovered that mental imagery has a significant positive effect on individual performance (Manz and Neck 2004; Manz and Sims 1980, 2001). Mental imagery can be useful when a problem stems from time pressure. In that case, the nurse manager would picture herself in a calm manner listening to the nurses' concerns over the

challenges at hand, offering timely encouragement, and providing useful, deliberate direction. Solutions can be created that can ultimately save time in the future.

When time pressure is at hand, having a deliberate strategy with self-leadership skills is key. While many factors in the surrounding environment can cause stress on a nurse manager, her/his state of mind is within her/his power. Dysfunctional thinking, however, often hinders the nurse manager in advancing a unit toward its preferred benchmarks (Goldsby et al. 2020). Fortunately, dysfunctional or self-limiting thinking can be changed to be more constructive with evidence-based self-leadership strategies.

A significant research finding in the past 30 years is that people can decide on the way they wish to think (Seligman 1991). In the book, Talking to Yourself, Dr. Pamela Butler proposes that people participate in "an ever-constant dialogue" with themselves so that they can pilot their behaviors, feelings, and even stress level (Butler 1983). Much of this self-dialogue is centered on where a person places their attention. Nurses often struggle over difficulties that are not within their power to change, such as situations resulting from time pressure. Many are burdened about consequences that they cannot anticipate. Then, when time pressure is at hand with the potential stress that comes with it, self-defeating thinking can be the cause of extra burden. Significant challenges stem from dysfunctional thinking patterns. The most common dysfunctional thinking patterns are (Manz 1992):

1. All-or-nothing thinking—one perceives issues as "black-and-white" instead of as complex situations with a lot of variables and possible perspectives (for example, if events do not play out as hoped, one distinguishes only all-embracing failure).
2. Overgeneralization—one oversimplifies a specific failure as having a perpetual nature to it (for example, a person may say to themselves, "I always screw up!").
3. Mental Filtering—one perseverates on one dissatisfying feature of something, thus misrepresenting all other aspects of reality (for example, a nurse manager may have one nurse in the unit who is particularly challenging to her/him, and she/he may think, "My employees all hate me!").
4. Disqualifying the positive—one disregards valuable occurrences (for example, "Well, I got lucky there. That will never happen again.").
5. Jumping to conclusions—one assumes certain conditions of a situation are negative before there is enough evidence to do so (for example, "The top administrators of the hospital are coming today to inspect the unit. They're bound to find something they're not happy with.").
6. Magnifying and minimizing—one heightens the significance of negative elements and lessens the presence of positive ones (for example, "Yes, the new nurses on the unit are doing great work, but you know they'll move onto higher paying hospitals. The good ones always do.").
7. Emotional reasoning—one is steered by negative emotions (for example, on entering the hospital, the nurse manager says to herself, "Well, I wonder what disaster will happen today on the unit.").
8. Labeling and mislabeling—one spontaneously applies undesirable labels to describe oneself, others, or an event (for example, during a break, the nurse manager sarcastically thinks to himself, "How did I end up being the king on this 'island of misfits'?").
9. Personalization—one accuses oneself for undesirable situations or conclusions that have other origins (for example, "I just know these new directives from the director are because of something I did wrong!").

Psychologists point to cognitive distortions as sources of these mental states that can undercut personal effectiveness (Neck and Barnard 1996). Even forms of depression can be the result of these mindsets. When nurse managers can recognize their self-defeating self-talk when it is taking place, they have the opportunity to alter and re-verbalize these personal dialogues. There is always potential for creating a more positive outlook that will enhance their performance and satisfaction (Goldsby et al. 2020).

Once the self-leadership practices of the nurse manager and nurses are improved, the interactions between the two parties can be honed as well. Social cognitive theory (SCT) (Bandura 1986)—the underlying theoretical foundation of self-leadership—explains that performance is the outcome of a three-way relationship between a person's thoughts, actions, and surroundings in which they find themselves. Self-efficacy, which is a self-assessment of a person's ability to achieve specific undertakings, is in particular an important concept of social cognitive theory. Thus, self-efficacy is also significant within the practice of self-leadership (Neck and Houghton 2006). Furthermore, a chief aspiration of self-leadership practices, including thought pattern strategies, is the development of high self-efficacy prior to performing an activity (e.g., Manz 1986; Manz and Neck 2004; Neck and Manz 1992, 1996a). Thus, increased task-specific self-efficacy promotes superior performance expectations (Bandura 1991). Backed by empirical research, self-leadership has been found to be a very helpful process for achieving perceptions of high self-efficacy and task performance (Neck and Houghton 2006). According to self-leadership theory—to the degree that an activity or task is selected—a strong belief in self-determination coupled with the application of practiced skills in increasing a sense of proficiency can enhance a person's performance on a task (Neck and Houghton 2006). In other words, once a person truly believes something is within their hands to do and that they have the ability to do it, they have a much better chance of doing so. The aforementioned self-leadership strategies intentionally practiced over time increases that desired self-efficacy.

It should be noted though that self-leadership is not an isolated process. Improving not only the personal habits of thoughts and behavior but the interactions between all parties in the environment is crucial as well. After all, much of time pressure can be due to systematic factors within a unit. Systems improvements require the involvement of the whole team. Once the nurse manager has improved her/his practices of self-leadership, it is time to improve the environment he or she co-exists in with others in the unit. Turning a manager's employees into better self-leaders themselves is a process known as SuperLeadership. The best managers set the example of what a good self-leader does and empowers and coaches the rest of the team to reach that same level of self-performance. When achieved, a team can outperform others who must wait for a manager to inform them how to handle complex situations. In a sense, the SuperLeader has inculcated the values and goals into each team member to exceed what she/he can do alone (Manz and Sims 1989, 1991; Manz 1990, 1991, 1992a; Neck and Houghton 2006). The best SuperLeaders of self-managing teams encourage and support their employees to learn and practice the self-leadership process (Neck and Houghton 2006). The nurse manager, after all, cannot find or fix all the factors and issues causing time pressure on the unit.

Given this extensive review of research on self-leadership theory and practices for better performance, we offer the following metaconjecture:

> **Metaconjecture 3:** *In situations where nurse managers face increased time pressure, proper application of self-leadership practices can improve psychosocial care; i.e., self-leadership practices positively moderate the negative relationship between time pressure and psychosocial care.*

## 4. Discussion

Thus far, this paper has provided a phenomenon of concern to nurse managers. Specifically, we discuss the negative impact of time pressure on psychosocial care, which is a key performance outcome for healthcare, and offer two evidence-based approaches to positively moderate that negative relationship. We have also provided metaconjectures to better demonstrate how practitioners can deliberately manage their time pressure situations. Yet, research tradition alone may not fully demonstrate the impact the application of theory to practice can have in a nurse manager's work environment. In this section, we go a step further to demonstrate how the quality of interactions between nurse manager and staff can be improved, and thus lead to a better climate for patient care. Therefore, we now provide a scenario to better understand how the Time Pressure Mitigation Model for Nurse Managers might apply to decision making in a healthcare environment:

Tina was a young nurse who found herself in a challenging situation on a medical/surgical unit. Currently, she was responsible for an elderly patient who was upset because his pain medication was not sufficiently relieving his discomfort. Visiting family members were also giving Tina a challenging time with regular interruptions, unhappy that their father had pushed the call light repeatedly and felt he was not receiving the attention he deserved from his nurse. It seemed the whole day had gone like this for Tina. Tina started her day with documentation being behind from the previous shift, and she had two new admissions waiting for her in the Emergency Department. Additionally, the phone was ringing intermittently due to the nursing secretary calling in sick with no replacement at hand at that time. This was a stressor that she felt needed attention by someone other than herself. Her patient load was filled with several patients with high acuity who needed a variety of treatments this day. She felt that she was trying her best with the time that she had with this patient who was in pain, but she felt overwhelmed. Unfortunately, the next time Tina went by the room of the patient that was in pain, she blew up and shouted into the room, "Look. I don't have time for this! Your dad will get his needs met, but I've got two other rooms I have to deal with right now that have patients in a lot worse condition who need my attention!"

Suzanne, the nurse manager on the unit, hearing the commotion, entered the room to find the patient crying, an upset family pacing, and Tina dashing down the hallway. Suzanne reassured the family that she would help address the issue immediately, which she does with the rounding doctor at hand. Suzanne was not happy with Tina's interaction with the family, but unfortunately scenarios of this nature were far from uncommon on her unit. Time-pressure seemed to be a constant adversary for both her and her nursing team. She felt the whole team needed self-leadership training and at this point she needed to step up and practice some of her own self-leadership skills. Suzanne decides to use constructive self-talk to improve her interaction with the distraught patient and the family members, as well as Tina, who seemed to be in crisis mode. She says to herself, "I've dealt with similar problems in the past, and the stress on my unit has been lessened." "I picture myself in a calm manner listening to the patient and family and providing individualized care." She also addressed Tina's concerns over her challenges and offers her timely help, encouragement, and useful, deliberate direction. She also visualizes a later meeting with Tina going well and leaving with a feeling like she can tackle her next shift in a positive manner. She tells herself she is in this role because of how she's handled similar situations. Suzanne reminds herself that, though her unit is on the verge of being understaffed, she has just hired a new nurse. That hire will surely help spread some of the duties among the staff. Then Suzanne returns to the patient that was in pain to check on his status.

In the above situation, we observe a nurse manager addressing a poor decision made by a frontline nurse under time pressure. In the hypothetical scenario, self-leadership practices help her envision a way to provide better psychosocial care in her unit. Time pressure situations like this could also be moderated with better time management practices as well. For Suzanne to provide improved psychosocial care to her patients in her unit, the Time Pressure Mitigation Model for Nurse Managers suggests that she become an expert implementer and coach of time management and self-leadership practices.

## 5. Conclusions

This conceptual paper examined the research question: How can nurse managers and nurses provide quality patient care by making better decisions under time pressure? Many factors contribute to nurse managers experiencing time pressure, such as: (1) patient safety; (2) patient satisfaction; (3) hiring; (4) staffing; (5) up-to-date, required education of the nurses; (6) playing the role of a front line nurse when the unit is understaffed; (7) serving as the liaison go-between, communicating changes on new policies or procedures that come from the administration and keeping up with Joint Commission on Hospital Accreditation requirements; (8) serving as the liaison go-between of the front line nurses and the hospital administration; (9) keeping doctors satisfied; (10) dealing with higher patient acuity; and (11) addressing the high stress among nurses, which includes burnout, suicide, and substance abuse (Davidson et al. 2018). Given this list of responsibilities, it is understandable why nurse managers are also described as nurse executives. In

that, they have responsibilities in line with someone with a business degree requiring knowledge on financial management, sourcing and procurement, operations, and other business skills that must be learned on the job, all while meeting their patient and nurse responsibilities (Roussel et al. 2020). Therefore, readers of this paper who are not healthcare providers but who are in similarly time-pressured situations may find the model and suggested practices useful as well.

As with other professions that experience significant time pressure, handling an amalgam of such diverse concerns can cause emotional exhaustion and often lead to burnout for the nurse manager (Warshawsky and Havens 2014). However, we contend that the reason nurse managers incur such consequences is because they carry much of the load of their units in providing quality care and meeting administrative objectives. Operating as the center of the unit rather than as a facilitator of their team's growth and development can hinder deliberative improvement of their processes and practices. As a result, they may find themselves running from one crisis to another. The nurses on the unit also experience this time pressure, as nurse managers often do not model appropriate time management skills and operational efficiencies and improvements. However, it is our hope that by applying the practices of time management and self-leadership encompassed in the Time Pressure Mitigation Model for Nurse Managers, more interactions with a nurse manager's staff will be positive and productive.

Helping nurses to become better time managers should be a part of each nurse manager's day with their staff. Deliberate and persistent use of time management and self-leadership practices will also reduce the dissatisfiers that drain time and energy on the unit and open up more opportunities to find the little fixes that add marginal time savings as well. Workshops and on-the-job training implementing evidence-based approaches in time management (Waterworth 2003) and self-leadership (Neck et al. 2019) would provide structure in reaching these objectives. As Waterworth (2003, p. 433) observed, "Literature on time management in nursing is mainly anecdotal", without providing structure regarding what specific strategies to use and when. Additionally, with regard to self-leadership, practitioners may confuse this research tradition with popular books that fall into the commercial category McGee calls "Self-Help, Inc." (McGee 2005). However, self-leadership, like time management, has proponents in academia as well as popular culture, and it is the evidence-based practices that our model advises nurse managers to apply. Application of anecdotal work alone can lead to hit-and-miss results, which may lead practitioners to abandon a focus on better time management and self-leadership, and fall back into their past habits. The importance of a model like the one proposed in this article is that it is comprised of key variables related to a phenomenon of concern (time pressure and its effect on psychosocial care), evidenced-based practices grounded in extensive research traditions (time management and self-leadership), and explanation for how the variables create the potential for better performance (the negative relationship between time pressure and psychosocial care and the positive moderating effects time management and self-leadership practices have on that relationship). As Lewin stated (Lewin 1952, p. 169), "There is nothing more practical than a good theory." A good theory is based on many studies of a phenomenon of concern that over time leads to better practices for progress in a field. Thus, it is our hope that nurse managers who apply practices in a research-based model as the one proposed in this conceptual paper may find their pursuit of better psychosocial care in their units to be met with more efficiency and effectiveness. Before long, the unit will find it has more time to focus on the work they care about and be in a less stressed state of mind for handling the crises that do inevitably arise.

In a qualitative study of nursing, nurse managers summarized the complexity of their workplaces: "Keeping everyone happy—the docs, the administration, the patients, the nurses". "I'm a staff morale-booster, the problem solver, and counselor, and anything else I can think of to keep people relatively happy" (Tuckett et al. 2015). The breadth and scope of nurse manager roles can be tremendous. However, because front-line nurse managers are "close to the action," they are in paramount positions to foster change in creating a positive work environment for nurses that parleys the emergent pressures that are commonplace. Many organizations place their managers in similar situations as nurse managers when they are confronted with many of the same challenges

found in today's healthcare environments where time pressure is a common occurrence. Another common managerial issue is the promotion of high-performing front-line employees into management positions. What worked for the manager as an individual performer will not necessarily be effective for the frontline they now oversee. With regard to nurse managers, many are promoted from front-line nursing status without master's degrees or leadership certifications that could have prepared them for the complexities of the position (Mathena 2002). While the tasks and responsibilities of the position can be learned fairly quickly, leadership is a skill that takes time to hone. However, learning self-leadership skills can have incredible results if consistently applied with employees.

It is our hope that this conceptual paper sheds light on the dynamics of these situations, and provides helpful advice for improving decision making, reducing time pressure and its related stressors, and bettering the healthcare experience. Metatriangulation with thorough research traditions in time pressure, psychosocial care, time management, and self-leadership provide support that the Time Pressure Mitigation Model for Nurse Managers warrants application in hospital units. But support for the model would be enhanced by empirical research. A common practice in emergency departments that requires sound decision making in often stressful situations is triage. Triage places a registered nurse in a situation to assess the treatment acuity—that is, how long a patient should be allowed to wait for needed medical care—and decide which interventions are required for emergency care in the moment. Nurse managers hold significant responsibility for the outcomes of triage in their units. For future study to further support the Time Pressure Mitigation Model, researchers could test whether time management and self-leadership practices improve psychosocial care in units with similar time pressure as emergency departments face. Given the literature on triage practices and decision making though (Mackway-Jones et al. 1997; Gerdtz and Bucknall 1999, 2001; Cone and Murray 2002; Barberà-Mariné et al. 2019), triage decisions in an emergency department may be a good starting point for future empirical research. Yet, the model in this paper is not limited to nurse managers. As warranted by metatriangulation, models based on thorough research can be applied to different domains as well. In most cases, extensions of the model in this paper could be achieved by replacing psychosocial care with another outcome variable. For example, hospitality managers often face time pressure serving large numbers of guests during busy periods such as hosting conferences and conventions. In a hotel setting, guest satisfaction could be measured as the dependent variable pre- and post-test to time management and self-leadership training. Other fields could perform similar experiments by modifying the performance outcome. We contend that the model in this conceptual paper provides one route to improving decision making for a wide range of leaders in time pressured environments.

**Author Contributions:** Conceptualization, E.G., M.G., C.B.N. and C.P.N.; Writing—original draft, E.G. and M.G.; Writing—review & editing, C.B.N. and C.P.N.

**Funding:** This research received no external funding.

**Conflicts of Interest:** The authors declare no conflict of interest.

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
