# Peer review of "Under Pressure: Time Management, Self-Leadership, and the Nurse Manager"

_admsci, doi:10.3390/admsci10030038_

Round 1
Reviewer 1 Report
Dear Author(s), I appreciate the effort of writing this paper, but it seems quite unstructured, and therefore, difficult to evaluate. I can't see any research question, any literature review (and criteria with review has been produced). There are no words about the method, and the case about "Tina and Suzanne with time pressure". I would like to suggest you to submit again the article, but starting with a more structured approach the highlight research question: i.e. how can nurses decide effectively under pressure context?, the case method: i.e. case study, ethnograpic research, and so on, literature review i.e. analyze the literature in a critical fashion by considering the decision making process on healthcare sectore under pressure context and so on. This could make your research more scientific and understandable.
I hope you can find these information useful for the future submissions. Good luck.
Reviewer 2 Report
The topic area is an interesting one however the work need strengthening through:
updating key literatures
contextualising the study
identifying the key disciplines and theoretical perspectives impacting the study
outlining the key conceptual frameworks and the relationship between them
including evidence of research design and a justification around the data presentation
clear statement of contribution to theory and practice
Reviewer 3 Report
Dear Author(s),
I would like to thank you for the opportunity to read your manuscript “Under Pressure: Time Management, Self-Leadership, and the Nurse Manager” for Administrative Sciences. I found it in line with the interest of the journal and Special Issue. I have read the paper a several times before writing this report and I think that the paper is interesting in its scope but it needs major revisions, especially in its structure. Comments are the following:
- Structure:
- Please, within the abstract refer to the type of paper that readers are going to read, conceptual? Moreover, it is too late referring to self-leadership theory at the end of the abstract. Please, put it before. Within the last line of the abstract readers expect to know the ‘results’ and/or ‘implications’ of your paper;
- Introduction: I have well understood the aim of your paper and I really comment on going on with this line of research. However, the introduction should be revised as to better fit the readers of the journal and give the necessary info that are expected from an intro. In particular, at the beginning, initially refer to management decision makers in general and their increasingly need to manage time pressure in their jobs. In doing that, I really encourage you to read the following:
Hambrick, D. C., Finkelstein, S., & Mooney, A. C. (2005). Executive job demands: New insights for explaining strategic decisions and leader behaviors. Academy of management review, 30(3), 472-491;
Ganster, D. C. (2005). Executive job demands: Suggestions from a stress and decision-making perspective. Academy of Management Review, 30(3), 492-502.
Then, you can make a parallelism with nurses’ management and say that some useful insights from the inter-relation of self-leadership literature and nurses literature. After having problematized literature, and you can mantain the part in which you explain what nurses’ literature has said, there should be a paragraph in which you better point out the “However,...”. Saying what it missing and what is the research question. Then, please, say that you will answer this question with a conceptual methodology (I give you more comments later). After that, please, refer to the implications of your work for management and nurses’ literature. I am going to suggest you also to report here the implications of a framework that you can propose;
- Literature: the first section of the literature seems to be very generic, I think that most of its contents can be shifted to the introduction as to introduce nurses’ management, while the others can be just cut.
- Methodology: to be built. Please, see my specific comments about that.
- Literature: Please, apart from the specific comments I am going to give you as to improve it, try to do the following: merge all the literature sections that are about nurses’ literature under one umbrella headline and do the same for the self-leadership literature. Doing that, you will have two theoretical sections that then should be speak each other in the results/discussion section;
- Results/discussion section: to be built. Please, see my specific comments about that.
- Conclusions: this section is to be a bit reshaped according to the following structure. Please, report the research question and its relevance for management and nurses literature. Report the methodology used. Please, sum-up in 5-10 lines the results/discussion section. Then say how these results complete: nurses literature, and self-leadership literature (recurring to the cited works in the theoretical background). Provide future research lines for theory, both for nurses and self-leadership literature. Provide implications for practice, both for nurses and self-leadership literature, starting, however, with the ones for managers and then to nurses as to give a stronger management focus to the paper.
- Methodology: Please, before starting with the parallelism of the nurses’ literature and self-leadership theory, build a methodology section in which you explain the methodology you adopted for building this conceptual work. I really suggest to look, to give you a good example, to the recent work of Cristofaro (2020) in which he built a conceptual work through a meta-triangulation approach – that really will work also for your case.
Cristofaro, M. (2020), “Unfolding Irrationality: How do Meaningful Coincidences Influence Management Decisions?” International Journal of Organizational Analysis, DOI: 10.1108/IJOA-01-2020-2010
You can find another good example of how this meta-triangulation works in practice for building conceptual papers in the following contributio. However, here there is no methodological section that explains how the paper has been built.
Abatecola, G. (2014), “Untangling self-reinforcing processes in managerial decision making. Co-evolving heuristics?”, Management Decision, Vol. 52 No. 5, pp. 934-949.
In practice, it is asked you to formalize your approach, that is quite right in its execution, thus proposing the nurses’ literature, then the self-leadership literature, and finally to let the two speaking each other.
- Relationship to Literature:
In general terms, I think that you really know nurse literature and self-leadership one. However, there are a series of papers that can give you further insights.
Nurses management and decision making:
- Aitken, L. M., Marshall, A. P., Elliott, R., & McKinley, S. (2009). Critical care nurses’ decision making: sedation assessment and management in intensive care. Journal of Clinical Nursing, 18(1), 36-45.
- Bucknall, T. K. (2000). Critical care nurses’ decision‐making activities in the natural clinical setting. Journal of clinical nursing, 9(1), 25-36.
- Cone, K. J., & Murray, R. (2002). Characteristics, insights, decision making, and preparation of ED triage nurses. Journal of emergency nursing, 28(5), 401-406;
- Gerdtz, M. F., & Bucknall, T. K. (2001). Triage nurses’ clinical decision making. An observational study of urgency assessment. Journal of advanced nursing, 35(4), 550-561;
- Gerdtz, M. F., & Bucknall, T. K. (1999). Why we do the things we do: applying clinical decision-making frameworks to triage practice. Accident and emergency nursing, 7(1), 50-57.
Self-leadership theory: within it I would suggest inserting here also some papers about management decision making that took insights from clinical settings:
- Rudolph, J. W., Morrison, J. B., & Carroll, J. S. (2009). The dynamics of action-oriented problem solving: Linking interpretation and choice. Academy of Management Review, 34(4), 733-756;
- Barberà-Mariné, M. G., Cannavacciuolo, L., Ippolito, A., Ponsiglione, C., & Zollo, G. (2019). The weight of organizational factors on heuristics. Management Decision.
Moreover, about the nurses literature, I found it a bit confusing. What I strongly encourage you to do is to find 4-5 points related to the decision making of nurses and trying to let them speak with the self-leadership literature. In one of the suggested papers of nurses management (Cone and Murray, 2002), for example, are considered: intuition, assessment abilities, good communication, and critical thinking. I am not saying that you have to take the same points, but I would like you, from the very beginning of the nurses management literature section to declare that you will structure your literature on some points and these last ones should be also treated in the self-leadership literature (so, also this one should be restructured according to these points).
- Results/Discussion:
At this moment you do not have a results/discussion section, I strongly encourage you to build it due to the fact that, right now, you have two literature sections that do not speak each other. So, build this section looking at the suggested works of Abatecola (2014) and Cristofaro (2020). The discussions sections of these works propose a field in which prior literature speak each other; in doing that, they propose a model/framework/table that describe a new look of the phenomena by taking used perspectives together. In better conveying ideas of this section you can approach ALSO with an anecdotal example, that you already have in your manuscript. This last one does not work alone in the way you propose it right now. It should be inserted in a discussion section accompanied by a model that comes out from the two literature fields speaking each other.
- Quality of Communication:
The quality of the English is good. There are some elements about the communication style that should be fixed:
- There are some double spaces around the text (e.g., third line of the introduction before ‘One’);
- Line 44 of page 1 there is a wrong citation: Bucknall, T.,
- Please, revise the way in which you cite because sometimes you put the ‘and’ while some others not. Please, refer to journals’ guidelines.
Comments to the Author
I generally think that your contribution is interesting and original for the decision making literature, but it needs a good reshaping of sections.
I wish you good luck with your research and I strongly hope that all my comments can be useful for its development.

Round 2
Reviewer 1 Report
Dear Author(s) the article seems improved even if I don't feel sure about the consistency of the case. I propose you to delete the case and leave only the theoretical framework as elicited in the previous part of the article.
The case is not sufficient to validate the theoretical framework, in my opinion.
However, this version of the article sounds good for the submission, and also the updating of literature is consistent.
Please correct the concept of meta conjecture (p.5): is meta conjecture or micro conjecture (p.6) ? there is something wrong in the article from this point of view.
Good Luck
Reviewer 2 Report
Paper much improved however the contribution could still be more clearly articulated - the `so what` question. Think this is a relatively easy fix at the end of the piece. Consider in more detail what your model proposes, why this is required and how you see it being articulated in practice. The latter is the most important element - how do you see the model being used and being of practical importance. This will really strengthen your contribution.
Reviewer 3 Report
Dear authors,
thank you for having given me the possibility to read the new version of the paper entitled “Under Pressure: Time Management, Self-Leadership, and the Nurse Manager” for Administrative Sciences’ Special Issue on Entrepreneurial and Managerial Decision Making.
I have read the paper and answers to reviewers several times before writing this comment and I really think that the paper has been really improved since the prior version. However, some minor revisions should be made as to recommend the ‘accept’ decision. In particular:
- Please, highlight all the improvements you have made in yellows or other colours rather than working with track changes. It is very difficult to understand what parts are new and what are not. This would make the duty of reviewers be easier.
- Abstract: please, spell out that this is a conceptual paper. Moreover, your abstract does not seem aligned with your intro. Indeed, within the abstract you are focusing your attention on nurse managers and nurses, while in the intro your main focus is on leadership. My suggestion is to give more weight on leadership within the abstract. Also in this case, as I will point out within the comment to the introduction, the abstract does not seem to face a research question or a topic that is for interest for decision makers. Judgment is cited with reference to patients....you have the same lack of focus on decision making just before the ‘Insert figure 1’ when you speak the aim of the model.
- Introduction: in the first 5-6 lines there is no citation inserted, you should support you statement. Moreover, please, make a distinction of what is a manager and who is a leader. You use these terms in an interchangeable way while they are not. Be consistent throughout the text. Then, before the research question there should be, following the prior sentence, an ‘However’, so pointing out what others have not done. Yet, the research question is disconnected with the aim of the Special Issue, which is on entrepreneurial and managerial decision making. Please, try to rephrase it (reviewer 2 gave you some suggestions in this direction). Last but not least, I have pointed out in the prior round of review that you should spell out within the intro and after the RQ that you are proposing a conceptual paper and how it has been built (meta-triangulation and which kinds of literature you used to do microconjectures).
- Reference style: please, be consistent with journal’s guidelines. Sometimes you insert ‘et al.’ (i.e., line 53 page 2) while others you spell out all authors’ names;
- Line 345 and on: you have a list of items but its editing is confusing; please, insert numbers for them. The same for 479 and on;
- Discussion section: the example is nice and I have understood it, but this section should serve also as a discussion section – then you have conclusions. So, by maintaining the example, I strongly ask you providing references to the literature and connections between your scenario and the literature.
- Conclusions: the initial part of the conclusions seems ok but, as you can see by yourself, you initially start with a recap, then future research and then you have a list of nurses’ challenges. It does not work. In sum, you have merged the old conclusions with the new one without any concern about the liason to make. The old part, moreover, is full of theory, while a conclusion sections should point (as reported in the prior round of reviews): research question, recap of methodology, recap of results, future research directions, implications for practice, limitations.
In general terms, I have really liked the reshape you made on the manuscript, it has been an hard work. However, I am requesting you the last effort as to let the paper be more readable and adherent to the aim of the special issue: decision making.
